Paleoneuroanatomy of the European lambeosaurine dinosaur Arenysaurus ardevoli

Cruzado-Caballero P 1 2 pccaballero@unrn.edu.ar
Fortuny J 3 4
Llacer S 3
Canudo JI 2
1 CONICET—Instituto de Investigación en Paleobiología y Geología, Universidad Nacional de Río Negro , Roca, Río Negro , Argentina
2 Área de Paleontología, Facultad de Ciencias, Universidad de Zaragoza , C/Pedro Cerbuna, Zaragoza , Spain
3 Institut Català de Paleontologia Miquel Crusafont, C/Escola Industrial , Sabadell , Spain
4 Departament de Resistència de Materials i Estructures a l’Enginyeria, Universitat Politècnica de Catalunya , Terrassa , Spain
Farke Andrew
Electronic publication date: 2015 Feb 24
Publication date: 2015
Volume: 3
Electronic Location ID: e802
Received 2014 Nov 5; Accepted 2015 Feb 5
Copyright: © 2015 Cruzado-Caballero et al.
Copyright year: 2015
Copyright holder: Cruzado-Caballero et al.
License: This is an open access article distributed under the terms of the Creative Commons Attribution License, which permits unrestricted use, distribution, reproduction and adaptation in any medium and for any purpose provided that it is properly attributed. For attribution, the original author(s), title, publication source (PeerJ) and either DOI or URL of the article must be cited.
License URL: https://creativecommons.org/licenses/by/4.0/

Keywords: European lambeosaurine, Paleoneurology, Hadrosaurid, Paleobiology, Inner ear, Dinosauria

Funding: Instituto de Estudios Altoaragoneses Government of Aragón University of Zaragoza 221-393 Ministerio de Ciencia, Tecnología e Innovación Productiva Consejo Nacional de Investigaciones Científicas y Técnicas (CONICET) of the Government of Argentina This research has been supported by a grant from the Instituto de Estudios Altoaragoneses (Huesca) to the first author in 2009 to perform the CT scans. This research has been supported by the Government of Aragón (“Grupos Consolidados” H54, and “Dirección General de Patrimonio Cultural”) and the University of Zaragoza (221-393). P Cruzado-Caballero is supported by a postdoctoral grant from the Ministerio de Ciencia, Tecnología e Innovación Productiva Consejo Nacional de Investigaciones Científicas y Técnicas (CONICET) of the Government of Argentina. The funders had no role in study design, data collection and analysis, decision to publish, or preparation of the manuscript.

==============================
The neuroanatomy of hadrosaurid dinosaurs is well known from North America and Asia. In Europe only a few cranial remains have been recovered that include the braincase. Arenysaurus is the first European endocast for which the paleoneuroanatomy has been studied. The resulting data have enabled us to draw ontogenetic, phylogenetic and functional inferences. Arenysaurus preserves the endocast and the inner ear. This cranial material was CT scanned, and a 3D-model was generated. The endocast morphology supports a general pattern for hadrosaurids with some characters that distinguish it to a subfamily level, such as a brain cavity that is anteroposteriorly shorter or the angle of the major axis of the cerebral hemisphere to the horizontal in lambeosaurines. Both these characters are present in the endocast of Arenysaurus. Osteological features indicate an adult ontogenetic stage, while some paleoneuroanatomical features are indicative of a subadult ontogenetic stage. It is hypothesized that the presence of puzzling mixture of characters that suggest different ontogenetic stages for this specimen may reflect some degree of dwarfism in Arenysaurus. Regarding the inner ear, its structure shows differences from the ornithopod clade with respect to the height of the semicircular canals. These differences could lead to a decrease in the compensatory movements of eyes and head, with important implications for the paleobiology and behavior of hadrosaurid taxa such as Edmontosaurus, Parasaurolophus and Arenysaurus. The endocranial morphology of European hadrosaurids sheds new light on the evolution of this group and may reflect the conditions in the archipelago where these animals lived during the Late Cretaceous.

Introduction

Hadrosaurids are the most abundant ornithopod dinosaurs from the Late Cretaceous of Laurasia, with a very complete record including ontogenetic series, mummies, eggs, ichnites, etc. (see Lull & Wright, 1942; Horner, Weishampel & Forster, 2004 for reviews). This rich record also includes natural cranial endocasts or complete skulls allowing the generation of silicone or latex rubber models of the endocast (Lambe, 1920; Gilmore, 1924; Ostrom, 1961; Serrano-Brañas et al., 2006; Lauters et al., 2013). The endocranial morphology of hadrosaurids has been studied since the first quarter of the 20th century (as in the case of Edmontosaurus regalis (Lambe, 1920) or Lambeosaurus Gilmore, 1924). Nowadays, non-invasive techniques such as CT scans shed new light on the paleoneurology of dinosaurs and other extinct taxa (Witmer et al., 2008; Evans, Ridgely & Witmer, 2009; Godefroit, Bolotsky & Lauters, 2012; Lautenschlager & Hübner, 2013). CT scan techniques are currently common in biology and paleontology as a way of obtaining digital models of inner regions as in the case of endocranial morphology, where these cavities may sometimes be filled by matrix. One of the great advantages of the CT scan is also that it makes it possible to access features without destroying the specimen (by contrast with very old methods) with minimum manipulation of the specimen and to create 3D models allowing manipulation or measurement without damage to the specimen. A CT scan allows a 3D visualization with a high or very high resolution, depending on the type of CT scan used and the goal of the study.

To date, endocranial morphology is mainly known from North American specimens (Lull & Wright, 1942; Ostrom, 1961; Hopson, 1979; Evans, Ridgely & Witmer, 2009; Farke et al., 2013) and to a lesser extent from Asian remains (Young, 1958; Saveliev, Alifanov & Bolotsky, 2012; Godefroit, Bolotsky & Lauters, 2012; Lauters et al., 2013), including isolated individuals and ontogenetic series. In Europe, however, the cranial record of hadrosaurids is very scarce, and no paleoneurological analyses have yet been performed. The European hadrosaurids with cranial material are Tethyshadros, Telmatosaurus and Arenysaurus (Nopcsa, 1900; Dalla Vecchia, 2009; Pereda-Suberbiola et al., 2009b). In the case of Telmatosaurus, a latex rubber model of poor quality was described historically (Nopcsa, 1900).

Arenysaurus forms part of the rich hadrosaurid fauna from the Iberian Peninsula, although cranial remains are scarce (Cruzado-Caballero, Pereda Suberbiola & Ruiz-Omeñaca, 2010; Cruzado-Caballero, Ruiz-Omeñaca & Canudo, 2010; Cruzado-Caballero et al., 2013; Prieto-Márquez et al., 2013). It was described by Pereda-Suberbiola et al. (2009b) as the first European lambeosaurine hadrosaurid preserving most of the cranial elements, including an almost complete and uncrushed braincase (Fig. 1). The Arenysaurus remains, together with other hadrosaurid and lambeosaurine material, helped to exchange the vision of a primitive European fauna for one that is more diverse, permitting osteological comparison with derived hadrosaurid faunas from North America and Asia, and studies of the phylogenetic relations between them (Company, Galobart & Gaete, 1998; Casanovas et al., 1999; Pereda-Suberbiola et al., 2009a; Cruzado-Caballero, Pereda Suberbiola & Ruiz-Omeñaca, 2010; Cruzado-Caballero, Ruiz-Omeñaca & Canudo, 2010; Cruzado-Caballero, 2012). Recently, Cruzado-Caballero et al. (2013) and Prieto-Márquez et al. (2013) have raised the possibility of a North American influence on the European lambeosaurine fauna.

Figure 1 A 3D reconstruction of the braincase of Arenysaurus ardevoli.

(A) Braincase opaque, (B) Semitransparent braincase with the brain cavity endocast opaque.

The main goals of the present paper are (1) to describe the first 3D endocast of a European hadrosaurid, (2) compare the neuroanatomy of the European hadrosaurids with the other Laurasian ones, and (3) provide new insights into the paleobiology of the lambeosaurines, for which there has up to now been a scarcity of information in comparison with hadrosaurines (Evans, Ridgely & Witmer, 2009; Lauters et al., 2013).

Material and Methods

Studied material: MPZ2008/1 (Fig. 1), skull remains of the holotype of the taxon Arenysaurus (Pereda-Suberbiola et al., 2009b). The remains are from the Blasi 3 locality in the town of Arén (Huesca province, NE Spain). Postcranial remains of Arenysaurus have also been recovered (see Cruzado-Caballero et al., 2013).

Computed tomography: The cranial material of Arenysaurus was CT scanned at the “Laboratorio de Evolución Humana” (LEH) of the Universidad de Burgos (Spain) using an industrial CT scanner, the Yxlon Compact (Yxlon Compact; YXLON International; Hamburg, Germany). The braincase is broken into two pieces (one including the frontal, parietal, left postorbital and left squamosal while the other includes the right postorbital and right squamosal), and these were scanned separately. In both cases, the material was scanned at 200 kV and 2.8 mA and an output of 1024 × 1024 pixels per slice, with an inter-slice space of 0.3 mm. In the part of the skull with the frontal, parietal, left postorbital and left squamosal, there were 543 slices, providing an in-plane pixel size of 0.24 mm, while in the other part including the right postorbital and right squamosal there were 582 slices, providing an in-plane pixel size of 0.2 mm. Due to the density of the bone and internal matrix, the CT images present several artifacts such as beam hardening, cupping artifacts and ring artifacts. These artifacts made automatic thresholding impossible, because the grey pixel value changes. For example, the beam hardening artifact makes the edge of the object brighter than the center, and ring artifacts produce bighting and dark concentric circles. Furthermore, the grey levels of regions of interest are very similar to those of matrix regions. Therefore, the endocast segmentation was done manually. The segmentation was done in the 3D Virtual Lab of the Institut Català de Paleontologia using Avizo 7.1 (VSG, Germany), generating a 3D mesh of each CT scan. After the segmentation, the two 3D surfaces were united using the same software and looking for contact points in the 3D braincase surfaces. When these were perfectly fitted on the inside, the 3D endocast fitted too. Then digital measurements, including the volume, were obtained using Rhinoceros 4.0 and ImageJ.

Repository of the ct-data sets: Figshare http://dx.doi.org/10.6084/m9.figshare.1287781, http://dx.doi.org/10.6084/m9.figshare.1287779.

Cranial endocast

The braincase of Arenysaurus is almost complete, and the individual bones are heavily co-ossified (Fig. 2, see Video S1). It has a slight lateral taphonomic deformation that does not affect the validity of the three-dimensional digital model (see osteological description in Pereda-Suberbiola et al., 2009b). By means of the CT scan, an almost complete three-dimensional endocast has been reconstructed. The structures on the left side of the endocast are well preserved and have been digitally rendered, while those on the right side are poorly preserved and in some cases unable to be reconstructed. As a whole, it is possible to observe the incomplete olfactory bulbs, the cerebral hemisphere, cerebellum, beginning of the medulla oblongata, pituitary (hypophyseal) fossa, inner ear and the canal for almost every nerve from II to XII (Fig. 2).

Figure 2 Cranial endocast.

(A) right lateral, (B) left lateral, (C) dorsal, (D) ventral, and (E) anterior views. Abbreviations: car, cerebral carotid artery canal; cer, cerebral hemisphere; cll, cerebellum; dp, dural peak; ie, inner ear; mo, medulla oblongata; ob, olfactory bulbs; pit; pituitary fossa; ts, transverse sinus; vls, ventral longitudinal sinus. II–XII, nerves; II, optic nerve; III, oculomotor nerve; V, trigeminal nerve; V1, ophthalmic branch of nerve V; g V, trigeminal ganglion of nerve V; VI, abducens nerve; VII, facial nerve; VIII, vestibulocochlear nerve; IX, glossopharyngeal nerve; X, vagus nerve; XI, accessory nerve; XII, hypoglossal nerve.

The Arenysaurus endocast, as is typical in hadrosaurids, is elongate anteroposteriorly with an anteroposterior length of 116.5 mm from the base of the olfactory tract to the caudal branch of the hypoglossal nerve. The maximum width across the cerebral hemisphere is 48.4 mm, and the estimated volume of the endocast (including the olfactory bulbs) is 126.2 cm3. The total volume of the cerebral hemisphere is 65.4 cm3, comprising 53.3% of the total endocranial volume (excluding the olfactory bulbs). This volume value is close to the results obtained by Saveliev, Alifanov & Bolotsky (2012) for the adult specimen of the lambeosaurine Amurosaurus AENM1/123 (see Table 1).

Table 1 Measurements of length and volume for complete brain cavity and various brain regions.

Measurements were obtained from Lambe (1920), Ostrom (1961), Evans, Ridgely & Witmer (2009), Saveliev, Alifanov & Bolotsky (2012), Farke et al. (2013) and Lauters et al. (2013), and for Arenysaurus they were calculated from the digital endocasts using digital segmentation in the Avizo 7.1 program.

Taxa	Ontogenetic state	Specimennumber	Total length endocast without olfactory bulbs (mm)	Maximum width of the cerebral hemisphere (mm)	Volume total of endocast without olfactory bulbs (cm3)	Cerebral hemispheres without olfactory bulbs (cm3)	% cerebral hemispheres volumen with respect total volume	Olfactory bulbs volumen (cm3)	
Lambeosaurus sp.	Juvenile	ROM 758	113.2	43	88.32	35.1	39.74	2.9	
Corythosaurus sp.	Juvenile	ROM 759	110.1*	46.5	91.7	41.6	45.36	6.2*	
Parasaurolophus sp.	Juvenile	RAM 14000	–	36*	–	–	–	–	
Corythosaurus sp.	Subadult	CMN 34825	142	44.7	134.2	51.1	38.08	11.2*	
Hypacrosaurus altispinus	Adult	ROM 702	204	63.2	275.9	117.5	42.59	14*	
Amurosaurus	Adult	AENM 1/123	230	72	370	210**	56.76**	–	
Amurosaurus	Adult	AENM 1/123	230	72	400	240**	60**	–	
Amurosaurus	Adult	IRSNB R 279	154	65	290	87	30	–	
Arenysaurus	Subadult-Adult	MPZ2008/1	116.48	48.38	122.8	65.42	53.27	3.44*	
Notes.

* Incomplete or stimate.

** Include the volume of the olfactory bubs.

– No data.

On the other hand, the Arenysaurus endocast is considerably constricted lateromedially at the cerebellum level, with a maximum width of 31.3 mm in this region, and slightly constricted at the medulla oblongata (26.3 mm). Unfortunately, the vallecula system, described in the anterior part of the endocast of other hadrosaurids, cannot be observed in Arenysaurus due to the hard matrix that covers this area.

The angle of the major axis of the cerebral hemisphere to the horizontal is close to 45° in the endocast. According to Evans, Ridgely & Witmer (2009), this high angle corresponds to a lambeosaurine shape as opposed to that of hadrosaurines and other ornithopods, where the cerebral hemisphere is positioned more horizontally (Hopson, 1979).

The angle of flexure between the cerebellum and the cerebral hemispehere is very small, close to 10°, revealing that in this respect the endocast is similar to previously described adult Laurasian lambeosaurines (e.g., Hypacrosaurus altispinus ROM 702, Amurosaurus riabinini IRSNB R 279, AENM nos. 1/232 and 1/240; Evans, Ridgely & Witmer, 2009; Saveliev, Alifanov & Bolotsky, 2012; Lauters et al., 2013). According to Giffin (1989), pontine flexures are virtually absent and the possession of a nearly straight endocranial cavity is derived for “iguanodontids” and hadrosaurids. Further, in lateral view the cerebral hemisphere is not very strongly arched, as is the case in adult lambeosaurines and unlike young individuals (e.g., Parasaurolophus sp. RAM 14000). These different angles are possibly a consequence of more strongly arched frontals in young individuals (Farke et al., 2013). In Arenysaurus the angle of the dural peak is close to 114° (Lautenschlager & Hübner, 2013; Farke et al., 2013).

The olfactory bulbs are located anteroventral to the cerebral hemisphere; only the bases of the bulbs are preserved. It has not been possible to reconstruct them completely, because the skull is broken in the anterior part of the frontals. The left bulb is the more complete one, while the right bulb only preserves its ventral part. In anterior view, the left olfactory bulb has an inverted L-shaped morphology. In this view, it is also possible to observe that the left olfactory bulb is almost half the height of the cerebral hemisphere, as also occurs in the adult of Amurosaurus (IRSNB R 279, AENM nos. 1/232 and 1/240; Saveliev, Alifanov & Bolotsky, 2012; Lauters et al., 2013) and the subadult of Corythosaurus sp. (CMN 34825; Evans, Ridgely & Witmer, 2009). The olfactory bulbs are turned downward with an angle on the dorsal side of 127.6° (measured between the anterodorsal surface of the cerebral hemisphere and the dorsal surface of the olfactory bulb). The total volume of the partially preserved olfactory bulbs is 3.4 cm3.

Several authors have commented on the presence of vascular elements in endocasts (Osmólska, 2004; Evans, 2005; Evans, Ridgely & Witmer, 2009; Lauters et al., 2013). In the case of Arenysaurus, the transverse sinus can be seen on the lateral side of the cerebellum, and on the ventral side of the cerebellum and in part of the medulla oblongata the ventral longitudinal sinus can be discerned (Fig. 2).The Arenysaurus pituitary (or hypophyseal) fossa is located posteroventrally to the optic nerve. It is deformed on its left side. It has a length of 19.1 mm, a height of 32.8 mm, a width of 14.5 mm, and a volume of 3.6 cm3. The original volume of the pituitary fossa was probably bigger, but taphonomical deformation has caused a volume artifact. The size of the pituitary body appears relatively large, as in other hadrosaurids (Lauters et al., 2013). Posteroventrally, it is possible to observe the joining of two big cerebral carotid arteries (Fig. 2).

Cranial nerves

The canals for almost all the cranial nerves, excluding nerve I and IV, can be seen to be preserved on the left side. Through these canals other structures also accompanied the nerves (e.g., meninges, venous structures, arteries, etc.). The cranial nerves present the same configuration as in other hadrosaurids (see Hopson, 1979; Evans, Ridgely & Witmer, 2009).

Nerve II, or the optic nerve (CN II), is the most anterior nerve preserved. It is very small, tubular, and parallels the ventral side of the cerebral hemisphere (with a lateromedial width of 4.8 mm, and a dorsoventral height of 5.5 mm). It is located under the cerebral hemisphere and is joined to the pituitary anteriorly. This nerve is very small in comparison with hadrosaurids; for example, Hypacrosaurus (Evans, Ridgely & Witmer, 2009) and Amurosaurus (Lauters et al., 2013; Saveliev, Alifanov & Bolotsky, 2012). The optic chiasm can only be seen in left view and is represented by a low, rounded protrusion dorsal to the pituitary fossa. Nerve III, or the oculomotor nerve (CN III), is posterior to nerve II. It is located in the middle of the juncture between the pituitary and the midbrain. It is small and has a very short, tubular morphology (with a lateromedial width of 4.8 mm, a dorsoventral height of 6.5 mm and an anteroposterior length of 5.9 mm).

The next nerve preserved towards the posterior portion is nerve V, or the trigeminal nerve (CN V). From this nerve the ophthalmic branch (CN V 1) and the base of the trigeminal ganglion are preserved. However, the maxillary and mandibular branches (CN V 2−3) are not observed. The ophthalmic branch is 7 mm in height dorsoventrally and 2.4 mm in length anteroposteriorly.

The ventral side of the endocast preserves nerve VI, or the abducens nerves (CN VI). These are joined to the pituitary, and exits from it posteriorly to connect ventrally with the cerebellum. The nerves are flattened lateromedially and are wider than high.

Nerve VII, or the facial nerve (CN VII), is positioned anterior to the cochlea and near nerve VIII. This nerve is tube-like, very small and thin, with a slight widening dorsomedially on its distal side. It is ventral to nerve VIII and runs lateroposteriorly with respect to the anteroposterior axis of the endocast.

Nerve VIII, or the vestibulocochlear nerve (CN VIII), is dorsal to nerve VII. This nerve is only partially preserved, showing a very small portion of the base dorsoventrally flattened.

Nerve IX, or the glossopharyngeal nerve (CN IX), is posterior to the cochlea and runs laterally, touching the cochlea in its anteriormost part. This nerve is very slim in its basal part and is tubular in shape. At its lateral extreme the nerve is extremely expanded dorsomedially (3.1 mm) and lateromedially (3 mm).

Nerves X and XI, the vagus and accessory nerves respectively (CN X and XI), are joined and these possibly also join with a branch of nerve XII to form a single nerve. These joined nerves are very broad anteroposteriorly (6.8 mm) and are clearly lateroposteriorly directed.

Nerve XII, or the hypoglossal nerve (CN XII), is possibly formed by two branches. The more anterior branch could be joined with nerves X and XI. The second branch, which is more posterior presents an anteroposteriorly narrow base (2.2 mm) and a dorsoventral height (3.9 mm) that is expanded distally (with an anteroposterior width of 4.7 mm and a dorsoventral height of 5.58 mm). This nerve is only laterally directed.

Inner ear

The digital reconstruction of the inner ear is complete on the left side, whereas the right side just conserves part of the cochlea and the anterior and posterior semicircular canals. The general form of the inner ear is similar to that described in other hadrosaurids (Brown, 1914; Langston, 1960; Ostrom, 1961; Evans, Ridgely & Witmer, 2009; Farke et al., 2013), and, as discussed in Evans, Ridgely & Witmer (2009), it resembles the condition in extant crocodilians. The three semicircular canals are oriented in approximately the three planes of space, where the anterior semicircular canal is slightly higher dorsoventrally and longer (Fig. 3). This configuration is the most common one in vertebrates (Knoll et al., 2013). The arch of the anterior and lateral semicircular canals is circular in shape while the posterior semicircular canal is ellipsoidal. The anterior semicircular canal is slightly taller than the posterior semicircular canal (when the lateral canal is oriented horizontally). This difference between the dorsoventral heights of the canals is reflected in the ratio between them, which is 0.98 in Arenysaurus. With regard to their ampullae, the lateral ampulla is larger than the posterior ampulla and the anterior ampulla, as in Parasaurolophus sp. RAM 14000 (Farke et al., 2013) and unlike in Hypacrosaurus altispinus ROM 702 and Lambeosaurus sp. ROM 758 (Evans, Ridgely & Witmer, 2009), where the anterior ampulla is the largest, followed by the lateral ampulla. Moreover, in lateral view, the cochlea is boomerang-like, convex laterally and concave medially. In anterior view, it presents an S-shape with a sharp distal border and it has a length of 10.7 mm from the foramen vestibulea (Table 3).

Figure 3 Left inner ear.

(A) lateral, (B) anterior, (C) posterior, and (D) dorsal views. Abbreviations: asc, anterior semicircular canal; asca, ampulla of anterior semicircular canal; c, cochlear duct (= lagena); crc, crus communis; fv, fenestra vestibuli (= oval window); lsc, lateral semicircular canal; lsca, ampulla of lateral semicircular canal; psc, posterior semicircular canal; psca, ampulla of posterior semicircular canal; ve, vestibule of inner ear.

Discussion

The endocranial morphology among hadrosaurid dinosaurs is similar and characteristic of the family. Hadrosaurid endocranial possess a greatly inflated, smoothly rounded cerebrum, do not have a pontine flexure and the orientation of the cranial cavity within the skull is obliquely anterodorsal (Hopson, 1979). At a subfamily level (hadrosaurine-lambeosaurine) there are characters that can help to distinguish them, such as a brain cavity that is anteroposteriorly shorter or the angle of the major axis of the cerebral hemisphere to the horizontal in lambeosaurines (Evans, Ridgely & Witmer, 2009). Both characters are present in the endocast of Arenysaurus and are consistent with the lambeosaurine affinity of this taxon.

A previous paper (Pereda-Suberbiola et al., 2009b) considered that this Arenysaurus specimen belongs to a presumably adult individual on the basis of several osteological characteristics. The paleoneuroanatomical evidence supports this ontogenetic assignment, with the following features referred to adult hadrosaurid animals present in this specimen: an angle of flexure between the cerebellum and cerebral hemisphere that is very small as in lambeosaurine adults, as described by Evans, Ridgely & Witmer (2009), and the cranial sutures that are difficult to discern in the CT scan, as is usual in adult specimens.

However, some paleoneuroanatomical features herein reported are indicative of a subadult ontogenetic stage for this specimen: the total volume of the endocast without olfactory bulbs; the volume of the cerebral hemispheres without olfactory bulbs; the maximum width of the cerebral hemisphere (see Table 1). This puzzling mixture of characters from adult and subadult stages may reflect a possible first case of a certain degree of dwarfism evidenced by a hadrosaurid endocast. The hypothesis of a reduction in size due to insularism in European hadrosaurids has been proposed by several authors in the last decade and is supported by bone as well as track records (Vila et al., 2013 and references).

Moreover, Farke et al. (2013) have hypothesized that hadrosaurids such as the small ornithopod Dysalotosaurus lettowvorbecki present a dural peak (the angulation of the dorsal margin of the cerebellum, not its prominence) that is mostly unchanged through the ontogenetic stages. These authors suggest that the phylogenetic differences between the lambeosaurini and parasaurolophini tribes could be assessed in the light of the angle of the dural peak. In these terms, the lambeosaurins presented a wider angle (around 120°) while parasaurolophins presented a more acute angle (approximately 90°). We have observed hadrosaurins and lambeosaurins to display an angle of no less than 100°. In the case of Arenysaurus, this angle is approximately 114° (see Table 2). In sum, the angle of the dural peak may indeed be informative, suggesting that the condition with a greater angle could be a basal character and anangle less than 100° may be exclusive to the genus Parasaurolophus. Regarding the inner ear, although the general form is similar to the other hadrosaurids, it is possible to observe small differences in the semicircular canals with respect to the ornithopod clade (see Fig. 4). The anterior semicircular canal is tallest at the base of the clade (Dysalotosaurus and Iguanodon; the ratio of anterior/posterior semicircular canal height is 1.11 in Iguanodon), by contrast with some hadrosaurines, where the posterior semicircular canal is slightly taller than the others (Edmontosaurus; the ratio of anterior/posterior semicircular canal height is 0.92). Later, in the Lambeosaurinae subfamily, Parasaurolophus and Arenysaurus present anterior semicircular canals that are slightly taller (the ratio of anterior/posterior semicircular canal height is 0.97 in Parasaurolophus and 0.98 in Arenysaurus), while in the lambeosaurini tribe they are similar in proportions to those seen in Dysalotosaurus or Iguanodon (the ratio of anterior/posterior semicircular canal height is 1.58 in Hypacrosaurus and 1.16 in Lambeosaurus). In addition, Parasaurolophus and Arenysaurus share a lateral ampulla that is larger than the posterior and the anterior ampullae.

Figure 4 Endosseous labyrinths of the inner ears.

Endosseous labyrinths of the inner ears redrawn for: Dysalotosaurus, Lautenschlager & Hübner, (2013; Fig. 2(h)); Iguanodon, Norman, Witmer & Weishampel (2004; Fig. 19.9); Edmontosaurus, Ostrom; (1961; Fig. 59a); Lophorhothon, Langston (1960; Fig. 163a); Parasaurolophus, Farke et al. (2013; Fig. 16d); Hypacrosaurus and Lambeosaurus, Evans, Ridgely & Witmer; (2009; Fig. 8a,e) and Arenysaurus ardevoli, displayed on a cladogram redrawn from Horner, Weishampel & Forster (2004), with additional data from McDonald (2012) and Cruzado-Caballero et al. (2013). Left inner ear: Edmontosaurus, Arenysaurus, Hypacrosaurus and Lambeosaurus; right inner ear: Dysalotosaurus, Iguanodon, Lophorhothon and Parasaurolophus.

Table 2 Measurement of the angle of the dural peak for several hadrosaurines and lambeosaurines calculated from drawings and digital endocasts using ImageJ.

Measurements were obtained from the Arenysaurus endocast, Lambe (1920), Ostrom (1961), Evans, Ridgely & Witmer (2009), Saveliev, Alifanov & Bolotsky (2012), Farke et al. (2013) and Lauters et al. (2013).

Taxa	Angle of dural peak	
Edmontosaurus regalis
(N.M.C. No. 2289)	110.66	
Edmontosaurus
(A.M.N.H. No. 5236)	133.79	
Kritosaurus notabilis
(A.M.N.H. No. 5350)	132.28	
Corythosaurus sp.
(CMN 34825)	130.4	
Hypacrosaurus altispinus
(ROM 702)	139.08	
Lambeosaurus sp.
(ROM 758)	106.71	
Amurosaurus
(AENM 1/123)	123.77	
Amurosaurus
(IRSNB R 279)	138.56	
Arenysaurus
(MPZ2008/1)	117.08	
Parasaurolophus sp.
(RAM 14000)	90	

Table 3 The maximum length of the digital cochlea of Arenysaurus casts and of other lambeosaurines.

The maximum length of the digital cochlea of Arenysaurus casts determined using the Avizo 7.1 program, and of other lambeosaurines from Evans, Ridgely & Witmer (2009).

Taxa	Ontogenetic state	Specimen no.	Cochlea length (mm)	
Lambeosaurus sp.	Juvenile	ROM 758	9.2	
Corythosaurus sp.	Juvenile	ROM 759	11.9	
Parasaurolophus sp.	Juvenile	RAM 1400	7.6*	
Corythosaurus sp.	Subadult	CMN 34825	12.3	
Hypacrosaurus altispinus	Adult	ROM 702	16.7	
Arenysaurus	Subadult-Adult?	MPZ2008/1	10.72	
Notes.

* Not complete.

The vestibular system is involved in the coordination of movement, gaze control and balance, detecting head movement (sensing angular acceleration) in space and maintaining visual and postural stability (Paulina Carabajal et al., 2013). The morphology and size of the semicircular canals are related to locomotor agility and neck mobility and a decrease in the compensatory movements of eyes and head (see references in Knoll et al., 2013 and Paulina Carabajal, Carballido & Currie, 2014). According to Witmer et al. (2008), the reduction in the difference between the length of the anterior and posterior semicircular canals, and perhaps also in the height of these canals, may reflect a decrease in the compensatory movements of eyes and head in Edmontosaurus, Parasaurolophus and Arenysaurus. If true, this could be related with behavioral patterns that require less agility in the head movements (Sereno et al., 2007).

Likewise, we hypothesize that these differences in the vestibular system, i.e., the different ratios between the height of the anterior and posterior semicircular canal and the size of the ampullae, could be used as a phylogenetic signal to differentiate Edmontosaurus, Parasaurolophus and Arenysaurus from the rest of the hadrosaurids. However, more data are necessary to know the possible influences that these differences could have on phylogenetic interpretations or on behavior.

Conclusion

We provide the first complete 3D reconstruction of the brain cavity and inner ear of a European lambeosaurine, Arenysaurus. This cranial endocast presents the general pattern known for hadrosaurids and add to the record of hadrosaurid brain cavities from Laurasia. The osteological and paleoneuroanatomical data suggest that Arenysaurus was an adult individual that probably presented a certain degree of dwarfism due to insularity. Thus, Arenysaurus could be the first evidence of how dwarfism could affect hadrosaurid paleoneuroanatomy. Moreover, it presents an optic nerve that is unusually small, indeed very much smaller than that of other known hadrosaurid. Furthermore, the structure of the inner ear shows differences from the ornithopod clade with respect to the height of the semicircular canals. These differences can be explained principally in terms of a probable decrease in the compensatory movements of eyes and head, which would affect the paleobiology and behavior of these animals. We hypothesize that these differences in the vestibular system could be used as a phylogenetic signal.

Supplemental Information

Video S1 Video of the 3D reconstruction of the braincase of Arenysaurus ardevoli

Click here for additional data file.

We acknowledge the academic editor Dr. Andrew A. Farke (Raymond M. Alf Museum of Paleontology, Claremon, United States of America), Dr. Pascaline Lauters (Department of Palaeontology, Royal Belgian Institute of Natural Sciences, Bruxelles, Belgium and Service d’Anthropologie et Génétique Humaine, Université Libre de Bruxelles, Bruxelles, Belgium) and an anonymous reviewer for their comments on the manuscript. The authors sincerely thank Dr. Andrew A. Farke and Dr. Ariana Paulina Carabajal for valuable discussions, Dr. Elena Santos for the CT scanning, as well as Rupert Glasgow, who revised the translation of the text into English.

Institutional abbreviations

AEHM Amur Natural History Museum, of the Amur Complex Integrated Research Institute of the Far Eastern Branch of the Russian Academy of Sciences, Blagoveschensk, Russia (Amur KNII FEB RAS)

CMN Canadian Museum of Nature, Ottawa, Canada

IRSNB Institut Royal des Sciences Naturelles de Belgique, Brussels, Belgium

MPZ Museo de Ciencias Naturales de la Universidad de Zaragoza, Zaragoza, Spain

RAM Raymond M. Alf Museum of Paleontology, Claremont, California, USA

ROM Royal Ontario Museum, Toronto, Canada

Additional Information and Declarations

Competing Interests

Author Contributions

Data Deposition

The authors declare there are no competing interests.

P Cruzado-Caballero analyzed the data, wrote the paper, prepared figures and/or tables, reviewed drafts of the paper.

J Fortuny analyzed the data, contributed reagents/materials/analysis tools, wrote the paper, prepared figures and/or tables, reviewed drafts of the paper.

S Llacer contributed reagents/materials/analysis tools.

JI Canudo analyzed the data, wrote the paper, reviewed drafts of the paper.

The following information was supplied regarding the deposition of related data:

Figshare

http://dx.doi.org/10.6084/m9.figshare.1287781

http://dx.doi.org/10.6084/m9.figshare.1287779.

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
