# Peer review of "Paleoneuroanatomy of the European lambeosaurine dinosaur Arenysaurus ardevoli"

_PeerJ, doi:10.7717/peerj.802_

## Round 0.1 · original submission · Minor Revisions

The reviewers and I have noted a few areas that should be clarified or corrected in the revised manuscript. Perhaps the most pressing issue concerns the general statements about hadrosaur brain proportions / size as summarized in Table 4 and discussed in the text. As noted by Reviewer 1, if you choose to keep this in the revised manuscript, the analyses should be conducted in greater depth with perhaps a larger sample size. Perhaps a regression would be a better way to show the perceived differences of Arenysaurus?

OTHER COMMENTS FROM THE EDITOR:
Consider a minor title change...perhaps "Paleoneuroanatomy of the European hadrosaur dinosaur Arenysaurus ardevoli (Hadrosauridae)"...I'm not sure that the fact that it is European is relevant; the phylogenetic context is probably more important than geographic.
On the Video, "hyphophysis" should be spelled "hypophysis"
p. 6, given the ~0.2 mm pixel resolution as well as variation in how the data may be segmented (dependent upon threshold, etc.), I would recommend only providing measurements to the nearest millimeter. The same applies for other measurements--e.g., those on p. 7.
p. 6: Are the data going to be archived anywhere? This is important so that the results can be verified at a later time (or built upon).
p. 8: It is important to emphasize that you can only observe the canals for the nerves, not the nerves themselves. Other structures also accompanied the nerves--e.g., meninges, venous structures, arteries, etc. At a minimum, I would place a statement to this effect at this point in the text, as a reminder for the reader.
p. 8: "This joins the pituitary, which their exits from the posterior to connect ventrally with the cerebellum. " - this sentence is unclear
- Consider adding some comparisons to other endocasts - how do the nerve configurations described here compare with those of other published hadrosaur specimens?
p. 10: "With regard to their ampullae, the lateral ampulla is larger than the posterior ampulla and the anterior ampulla, as in Farke et al. (2013) and unlike in Evans et al. (2009) (where the anterior ampulla is the largest, followed by the lateral ampulla)." - Add a little more detail here, in particular mentioning which taxa are discussed in each paper. Can you also include measurements of the canals?
p. 12: "a lesser angle of 100º may be exclusive to the genus Parasaurolophus." - the only described Parasaurolophus endocast is a juvenile, so this also should be considered as a factor.
Table 1, Table 2, Table 4: RAM 14000, not RAM 1400 (confirm this is correct elsewhere in text, too)

Reviewer 1 ·

Basic reporting

The submitted manuscript includes all necessary sections and gives a good introduction on the background and the research field. Structure and formatting are clear. Figures are well made and clear.
The only minor issue I have is the language, which should be checked again by a native speaker. Some edits are included in the attached pdf.

Experimental design

The manuscript describes original research and the methodology is for most parts clearly stated. A few minor details should be included in the Materials and Methods section (see specific comments below) to make it more detailed reproducible.

Validity of the findings

Creation of the digital endocast and respective anatomical description are thorough and accurate. There are a few minor issues, with which I would disagree or where I would like to see clarification or more detail. Anatomical data and measurements are provided, but often not rigorously used to support the statements made. The discussion and conclusions are too speculative and often not supported by the data or even contradictory (see specific comments below).

Additional comments

The manuscript “Arenysaurus ardevoli, first paleoneuroanatomical description of a European hadrosaurid” describes the endocranial anatomy of a hadrosaurid specimen. The authors used a now common method of creating a digital endocast of the brain and inner ear structures and did a good job creating the digital model, which is not always the case for such reconstructions. The presented 3D reconstruction looks thorough and accurate. The accompanying description is, with some minor exceptions, solid and detailed and includes a number of measurements, which will facilitate the work of others in the future. I am disagreeing with some minor points and would expand the description in some sections, but apart from that the descriptive part is OK.
However, the biggest problem I have with this contribution is the discussion. While the authors took care to take various linear and volumetric measurements of the studied and comparative taxa, I often lack to see support for their conclusions (see points below). More measurements and (statistical) analyses will be required, if the authors wish to verify their hypothesis. A larger sample size of ornithischian dinosaurs (if available) should be included and the data more rigorously queried. Especially variation within the same species and between different ontogenetic stages should be checked carefully for significant trends. I am convinced several of the hypotheses can be supported, but at the moment, these are largely untested and not justified by the data.
Language and grammatical suggestions are included in the commented pdf, but I would recommend a native speaker to check the language before resubmission.
For this reason, I have to recommend major revisions for this manuscript. If further analyses are included and pending the revisions, I am sure this will make a valuable contribution.





Introduction

- Line 33: The endocranial cavity is not sorrounded by matrix, but the braincase bones. Matrix can sometimes fill the respective cavity. This should be clarified and the sentence rephrased.
- Line 32-34: I know what the authors mean to say here, but that doesn’t really come across for a more general audience. I would rather highlight (although only briefly) the possible benefits of digital endocast, such as high accuracy, the damage-free study of specimens, possibility to take measurements, etc.

Material and methods

- Line 77: What kind of scanner? CT, micro-CT, ...
- Line 86: “...hat made the segmentation of the different elements of the skull difficult, but not impossible.” Delete this part of the sentence, as it has no relevance to the study. Rather describe how these artefacts were tackled or how they affact the final digital model (automatic thresholding vs. Manual segmentation, loss of detail in specific regions, etc.)
- Line 89: “...were united using the same software...” How was that done? Were the tow individual surface models joined manually?
- Line 87-90: I would prefer some more information on the segmentation process here. For example, was the segementation perfomed manually, or could automatic thresholding be used?

Cranial endocast

- Line 92: “...and fused...” , do you mean the sutures are closed by that?
- Line 93: “...taphonomic lateral deformation...” Describe some of the taphonomic artefacts briefly (in about one sentence), as this will be important for the validity of the digital model.
- Line 132: “...inside-out L-shaped morphology...”, I don’t know how that would look like. Better rephrase
- Line 136: “...angle on the dorsal side of 127.6º”, in reference to the hemispheres or the horizontal plane?
- Line 140: “the opening of the caudal middle cerebral vein can be seen on the dorsal side of the cerebellum...”, looking at figure 2, I can find no inication for this vein. I would expect to see a paired opening/canal. The structure indicated in figure 2, merely shows the dural peak of the cerebellum.
- Line 140-142: “...and on the lateral side the dorsal head vein can be recognized (Figure 2). On the ventral side of the cerebellum at the beginning of the medulla oblonga, vascular elements can be made out.”, similar to the comment above, I can’t see any indication for the mentioned structures, neither are they labeled in figure 2.

Cranial nerves
- Line 153: The optice nerve canal is indeed very small in comparison to other endocasts. Are the authors confident about the identification? If correct this is noteworthy and should be highlighted more.
- Line 165: “This joins the pituitary, which their exits from the posterior to connect ventrally with the cerebellum.”, this sentence makes no sense, please rephrase.
- Line 169: Lateroposteriorly to what?
- Line 176-177: “...are separated attheir base, but then they join to form a single nerve.”, that’s very interesting and an unusual condition. Does that happen in other ornithischians endocasts?
- Throughout this section, I would avoid terms such as “tubular” or “button-like”. The digital casts of the nerves are representation of the respective canals, which are by their nature tubular. The button-like appearance is a result of a short canal. The different “morphologies” are simply an effect of the length of the canals and not necessarily characterisitic. If you want to describe this, rephrase the terms.


Inner ear

- Line 184: Strictly speaking, the vestibular apparatus encompasses just the semicircular canals and the vestibule, but not the cochlear. As the latter has been reconstructed consider rephrasing
- Line 188: I find the comparison with extant crocodilians not really appropriate, at least not without considering other ornithischian and saurischian dinosaurs first. Also, in what respect, does the inner ear resemble the condition in extant crocs. Further more as it is mentioned, the inner ear has a general morphology that is found in most vertebrates. I would like to see here more comparisons (and in more detail) with other ornithischian taxa, for which endocasts are available.
- Line 194: “...as in Farke et al. (2013) and unlike in Evans et al. (2009)...”, name the respective taxa here, especially as Evans et al. Have more than one taxon in their study.
- All in all, this section is rather short and undetailed in comparision to the previous sections. More comparisions and measurements could be included, for example the radius of the semicircular canals, etc.


Discussion

- Line 199-200: Simply mentioning that the morphology is characteristic is not enough. State the characteristics that support your statement. The reader shouldn’t have to glean the information from other papers.
- Line 206: What is meant by “sole” adult individual?
- Line 212-216: I very much doubt that endocast length is indicative of ontogenetic stage without a clear knowledge about variation throughout the clade. Yes, juvenile individuals will of course have smaller/shorter endocasts than adults, but if you don’t have an adult or juvenile specimen of the same species to compare with, this is largely speculation. Looking at table 1, I see a large variation (over 50%) of endocast length in three adult specimens of Amurosaurus. According to your statement, 1/123 IRSNB R would have to be a juvenile. If you really want to use endocast length, then compare a large number of known dinosaurian endocasts with skull length/basicranial length/etc. in order to find a statistically significant relationship.
- Line 218-220: Again, I find it difficult to believe based on average femur length and a possibly short endocast that this is a sign for beginning dwarfism. The listed samples in Tables 1-4 are not appropriate nor numerous enough to show any trend supporting this statement. While sample size is obviously a problem given the small number of endocast studies, there is still a problem that there are no juvenile/adult pairs of the same taxon to show a respective relationship. Also, is there any osteological indication for dwarfism in Arenysaurus?
- Line 226-233: The differences in the angle of the dural peak are intersting and I would like to see more testing in that direction. Why not include Dysalotosaurus and other ornithischian taxa, to test for phylogenetically informative differences?
- Line 235-240: Apart from Dysalotosaurus and Iguanodon, the differences between the anterior and the posterior semicircular canals are very small or even of equal height and I would be cautious to argue for taxonomically relevant differences in the more derived taxa considering taphonomic artefacts, etc. The authors have been very detailed with their measurements, but they are lacking here. A ratio of anterior/posterior semicircular canal height will show, if there is an actual trend. At the moment this has to be guessed qualitatively from the figures.
- Line 252-255. Again, I would be very cautious here. If, as mentioned in the previous sentences, the size differences of the semicircular canals are related to behaviour and function, then this makes a poor phylogenetic character.

Conclusions

- Generally, I think the conclusions should be considerably changed according to the aforementioned comments, however, some points below to consider.
- Line 265: Table 2 shows that the angulation in the same species and ontogenetic stage can be highly variable (Edmontosaurus, difference of 23 degree, Amurosaurus 15 degree). This highly suggests that there is a considerable degree of intraspecific variability and/or taphonomy involved. Therefore I highly doubt that this is an autapomorphy in Parasaurolophus without further data support.


Figures and tables

- Figures are well done and show all necessary detail. The only thing I would change is to make the individual figures of the endocast and inner ear larger, as there is still a lot of unused space between them.
- Table 1: What is the difference between cerebrum volume and cerebral hemispheres volume? Do you mean cerebellum volume?
- Captions for Tables 1 and 2 are wrong (same as Table 3)
- I don’t think the average values in Table 4 are appropriate or even significant. You are comparing a variety of different taxa, without even knowing the variation within one species.

Annotated reviews are not available for download in order to protect the identity of reviewers who chose to remain anonymous.

·

Basic reporting

The manuscript should be proofread by a native English speaker to enhance the intelligibility of the text.

Experimental design

No comments

Validity of the findings

The availability of the 3D reconstructions to other researchers is not clearly stated, see comment on the manuscript.

Additional comments

Please see attached pdf with minor comments on the manuscript.
The manuscript exposes interesting description of the first endocranial cast of an European hadrosaurid. It presents unpublished research and hypothesis and represents a valuable input on the neuroanatomy of hadrosaurid dinosaurs. The hypotheses about the inner ear anatomy and paleobiology of Arenysaurus are of interest.
The proposed manuscript is clear and well organized. The figures and references are relevant to the subject of the paper.
I recommend to extend the discussion on the endocranial morphology by calculating the REQ (reptile encephalization quotient). Values of REQ and cerebral volumes are available in the literature and could be compared with those obtained for Arenysaurus. It will give new input on the phylogeny and the paleobiology of hadrosaurid dinosaurs.

---

## Round 0.2 · Minor Revisions

Thank you for your revision of the manuscript in light of comments from the reviewers. Only a handful of issues remain - these are listed below. Once these are addressed, I should be able to issue a final decision in very short order.

p. 6: Are the data going to be archived anywhere? This is important so that the results can be verified at a later time (or built upon). If it is not possible to post the CT scans publicly, at the least they should be reposited at the institution where the specimen is held.

p. 8: It is important to emphasize that you can only observe the canals for the nerves, not the nerves themselves. Other structures also accompanied the nerves--e.g., meninges, venous structures, arteries, etc. At a minimum, I would place a statement to this effect at this point in the text, as a reminder for the reader.

Table 2: It is only necessary to present the angle of the dural peak to the nearest degree; anything beyond that is probably overly-precise.

---

## Round 0.3 · accepted · Accept

Thank you for your quick work on this! In proof, I would recommend two minor rewordings, neither of which requires a revision.

1) "Repository of the ct-data sets" change to "Repository of the CT data sets"

and

2) "Through these canals other structures also accompanied the nerves (e.g. meninges, venous structures, arteries, etc.)." change to "Through these canals other structures also accompanied the nerves (e.g. meninges, venous structures, arteries, etc.), but for the purposes of brevity the canals are discussed in context of the nerves alone." [or similar wording that would clarify the matter for readers]

I look forward to seeing this paper published soon!